# RARE EVENT PROBABILITY LEARNING BY NORMALIZING FLOWS

## ABSTRACT

A rare event is defined by a low probability of occurrence. Accurate estimation of such small probabilities is of utmost importance across diverse domains. Conventional Monte Carlo methods are inefficient, demanding an exorbitant number of samples to achieve reliable estimates. Inspired by the exact sampling capabilities of normalizing flows, we revisit this challenge and propose normalizing flow assisted importance sampling, termed NOFIS. NOFIS first learns a sequence of proposal distributions associated with predefined nested subset events by minimizing KL divergence losses. Next, it estimates the rare event probability by utilizing importance sampling in conjunction with the last proposal. The efficacy of our NOFIS method is substantiated through comprehensive qualitative visualizations, affirming the optimality of the learned proposal distribution, as well as a series of quantitative experiments encompassing 10 distinct test cases, which highlight NOFIS's superiority over baseline approaches.

## 1 INTRODUCTION

A rare event (Bucklew & Bucklew, 2004) is characterized by an occurrence probability close to zero (e.g., less than $10^{-4}$). The estimation of such rare event probabilities is of significant interest across various domains, such as microelectronics (Kanj et al., 2006; Sun et al., 2015), aviation (Brooker, 2011; Ostroumov et al., 2020), healthcare (Cai et al., 2010; Zhao et al., 2018), environmental science (Frei & Schär, 2001; Ragone & Bouchet, 2021), and autonomous driving (O'Kelly et al., 2018), as it can help avert significant economic losses or catastrophic events. To understand its significance, imagine a manufacturing process with a probability of $10^{-4}$ introducing defects into drug vials. This could result in approximately 100 defective vials out of the $10^6$ produced, leading to significant financial losses and triggering a public health crisis. Conversely, if the probability is less than $10^{-9}$, all $10^6$ vials will have a high likelihood to be free of defects.

The Monte Carlo (MC) approach is widely recognized as inefficient for the rare event probability estimation problem (Dolecek et al., 2008; Sun et al., 2015; O'Kelly et al., 2018). For instance, when aiming to estimate a small probability such as $10^{-6}$, the MC method may require more than $10^8$ samples to achieve a relatively low estimation variance. However, gathering such a large number of samples can be unaffordable, as typically the data acquisition needs to invoke expensive computer simulations in a real-world application. In other words, beyond the pursuit of estimation accuracy, the efficiency of data sampling assumes a critical role as well. To confront this challenge—ensuring precise estimation within a data sample budget, various methods rooted in statistics were established from diverse domains (Au & Beck, 2001; Allen et al., 2009; Sun et al., 2015).

We posit that the recently popularized technique of normalizing flows (Dinh et al., 2014; 2016; Papamakarios et al., 2021) provides an unprecedented and highly efficient tool for rare event probability estimation. The elegance of applying it to this task is that normalizing flows impose a sequence of transformations to shift a base distribution to a desired target distribution, and we realize that this procedure could be adapted to reflect the learning of a sequence of *proposal distributions* associated with several *nested subset events* (Au & Beck, 2001). By setting the original rare event as the last subset event, the ultimate shifted distribution in the normalizing flow will be a good proposal distribution for the original rare event. Thus, this final proposal distribution can be combined with importance sampling to generate an accurate estimate of the original rare event probability. In a nutshell, our contributions in this paper include:

- We proposed an efficient rare event probability estimation technique, termed NOFIS, short for normalizing flow assisted importance sampling. Its key is to utilize a sequence of pre-defined nested subset events and successively learn the corresponding proposal distributions by minimizing KL divergence losses.

- We conducted extensive 2-D visualizations to justify the superior capability of NOFIS in recovering the theoretically optimal proposal distribution. Moreover, compared to six baseline methods across 10 test cases, NOFIS consistently demonstrates superior estimation accuracy using fewer data samples.

## 2 BACKGROUND

**Normalizing Flows.** Normalizing flows (NFs) are a family of generative models that enable the modeling and sampling of intricate probability distributions (Kobyzev et al., 2020; Papamakarios et al., 2021). They achieve this goal by transforming a simple base distribution into a complex distribution through a series of invertible and differentiable transformations. These transformations are trainable and typically implemented as deep neural networks. Careful design of the neural network architectures is essential to ensure tractable computation. Prominent examples of such architectures include NICE (Dinh et al., 2014), RealNVP (Dinh et al., 2016), IAF (Kingma et al., 2016), MAF (Papamakarios et al., 2017), and Glow (Kingma & Dhariwal, 2018), among others. NFs have gained increasing attention due to their successful applications in various domains, such as variational inference (Rezende & Mohamed, 2015; Kingma et al., 2016; Chen et al., 2020), image synthesis (Dinh et al., 2016; Kingma & Dhariwal, 2018; Lugmayr et al., 2020), density estimation (Papamakarios et al., 2017), and MC integration (Müller et al., 2019; Gao et al., 2020; Gabri'e et al., 2021). Recently, Arbel et al. (2021); de G. Matthews et al. (2022) propose combining NFs with sequential MC to sample from unnormalized densities, which shares a similar spirit with our approach.

**Rare Event Probability Estimation.** The literature on rare event probability estimation spans a wide range of domains, and the specific formulations of the problem may vary slightly depending on the domain's specifications. One widely used approach is importance sampling (Bucklew & Bucklew, 2004; Kanj et al., 2006; Shi et al., 2018). Importance sampling involves sampling from a proposal distribution and estimating the rare event probability through a weighted ratio. Its effectiveness heavily relies on the quality of the proposal distribution. Another influential approach is subset simulation (Au & Beck, 2001). Subset simulation involves constructing a series of nested subset events with progressively decreasing occurrence probabilities, with the last representing the original rare event of interest. The estimation of probabilities for the original rare events is then decomposed into a product of several conditional probabilities, which are calculated using the Metropolis-Hastings algorithm based on Markov chains. Other noteworthy approaches include but not limited to Wang-Landau algorithm, sequential MC (Del Moral et al., 2006), line sampling (Schuëller et al., 2004), forward flux sampling (Allen et al., 2009), and scaled-sigma sampling (Sun et al., 2015).

## 3 PROPOSED METHOD

In this paper, we focus on the rare event probability estimation problem defined by a tuple $\mathcal{F} = (p, \Omega)$, where $p(\cdot) \in \mathcal{P}^D$ represents a $D$-dimensional data generating distribution, and $\Omega \subseteq \mathbb{R}^D$ represents the integral region associated with the rare event. Without loss of any generality and for conciseness, we parametrize $\Omega = \{\mathbf{x} \in \mathbb{R}^D | g(\mathbf{x}) \leq 0\}$ by a characteristic function $g(\cdot) : \mathbb{R}^D \rightarrow \mathbb{R}$. Our primary interest is to estimate the rare event probability represented by the integral:

$$P_r = P[\Omega] = \int_\Omega p(\mathbf{x}) \, d\mathbf{x} = \int \mathbb{1}[\mathbf{x} \in \Omega] \, p(\mathbf{x}) \, d\mathbf{x} = \int_{g(\mathbf{x}) \leq 0} p(\mathbf{x}) \, d\mathbf{x}, \quad (1)$$

where $\mathbb{1}[\cdot]$ represents the indicator function. The challenge lies in that $P_r$ is exceptionally small (e.g., less than $10^{-4}$) due to either $\Omega$ having an extremely small volume, or its majority being concentrated in the tail of the distribution $p$. In our context, the distribution $p$ is easy to evaluate and sample from (Sun et al., 2015), often following a standard Gaussian distribution. [1] On the other

---

[1] When the distribution $p$ deviates from a Gaussian form, a Power transformation (Box & Cox, 1964; Yeo & Johnson, 2000) can be applied to construct $\mathbf{x}'$ following a standard Gaussian distribution $p'$, so that we could equivalently solve the problem $\mathcal{F}' = (p', \Omega')$.

hand, $\Omega$ is complicated and unknown in advance, while evaluating the function value $g(\cdot)$ requires running computationally expensive black-box computer simulations. Thus, the goal of rare event probability estimation is to accurately estimate $P_r$ with as few function calls to $g(\cdot)$ as possible.

The importance sampling (IS) approach introduces a proposal distribution $q(\cdot) \in \mathcal{P}^D$ and estimates $P_r$ by drawing $N_{\text{IS}}$ i.i.d. samples from the distribution $q$:

$$P_r^{\text{IS}} = \frac{1}{N_{\text{IS}}} \sum_{n=1}^{N_{\text{IS}}} \mathbb{1}[\mathbf{x}^n \in \Omega] \frac{p(\mathbf{x}^n)}{q(\mathbf{x}^n)}, \quad \mathbf{x}^n \sim q(\cdot) \tag{2}$$

It is evident that as long as the support of $q$ includes that of $p$, the IS estimator remains unbiased (i.e., $\mathbb{E}_q[P_r^{\text{IS}}] = P_r$). Additionally, simple derivations demonstrate that the proposal distribution:

$$q^\star(\mathbf{x}) \propto p(\mathbf{x})\mathbb{1}[\mathbf{x} \in \Omega] = \frac{1}{P[\Omega]} \cdot p(\mathbf{x})\mathbb{1}[\mathbf{x} \in \Omega] \tag{3}$$

is theoretically optimal, as it can result in a zero-variance unbiased estimator (Bucklew & Bucklew, 2004; Biondini, 2015). It is important to note that since $\Omega$ is defined by the computationally expensive characteristic function $g(\cdot)$, $q^\star(\mathbf{x})$ is unknown in practice, and furthermore, direct sampling from $q^\star(\cdot)$ might not be feasible. As a result, it is common to implement the IS method by limiting the range of consideration for $q(\cdot)$ to a parametrized distribution family $\mathcal{Q}$ that allows for exact sampling, such as a mixture of Gaussian distributions (Biondini, 2015).

NFs are ideal to compose the distribution family $\mathcal{Q}$, due to their great expressive power and the capability to do exact density evaluation and sampling. For later simplicity, we introduce the notation $\Omega_a = \{\mathbf{x} \in \mathbb{R}^D | g(\mathbf{x}) \le a\}$ for any $a \in \mathbb{R}$. Motivated by (Au & Beck, 2001), we start from $M$ nested subset events $\Omega_{a_1} \supsetneq \Omega_{a_2} \supsetneq \cdots \supsetneq \Omega_{a_M}$ with decreasing occurrence probabilities, which are induced by a strictly decreasing sequence $\{a_m\}_{m=1}^M$ satisfying $a_M = 0$, ensuring that $\Omega_{a_M} = \Omega_0 = \Omega$. We emphasize that the value of $M$ and the sequence $\{a_m\}_{m=1}^M$ are both hyper-parameters of our algorithm, and we defer the empirical rules for setting them to the end of this section. As shown in Figure 1, we exploit an NF model defined by a base distribution $q_0(\cdot)$, and $MK$ invertible and trainable transformations $\{\mathbf{f}_i(\cdot) = \mathbf{f}(\cdot; \boldsymbol{\theta}_i) : \mathbb{R}^D \to \mathbb{R}^D\}_{i=1}^{MK}$, where $\boldsymbol{\theta}_i$ represents the $i$-th learnable parameters. The NF model starts from a random variable $\mathbf{z}_0 \sim q_0(\cdot)$ on the left end, and repeatedly applies each function $\mathbf{f}_i$ according to $\mathbf{z}_{i+1} = \mathbf{f}_{i+1}(\mathbf{z}_i)$. For simplicity, we denote the distribution associated with the intermediate random variable $\mathbf{z}_i$ by $q_i \in \mathcal{P}^D$. According to the change of variable theorem and the inverse function theorem, we have:

$$q_{j+1}(\mathbf{z}_{j+1}) = q_j(\mathbf{z}_j) \left| \det\left( \frac{d\mathbf{z}_j}{d\mathbf{z}_{j+1}} \right) \right| = q_j(\mathbf{z}_j) \left| \det \mathbf{J}_{\mathbf{f}_{j+1}} \right|^{-1} \tag{4}$$

where $\det(\cdot)$ denotes the determinant of a square matrix, and $\mathbf{J}_{\mathbf{f}}$ represents the Jacobian matrix of function $\mathbf{f}$. Take the logarithm of both sides in Eq. (4) and sum it by varying index $j$, yielding:

$$\log q_i(\mathbf{z}_i) = \log q_0(\mathbf{z}_0) - \sum_{j=1}^{i} \log |\det \mathbf{J}_{\mathbf{f}_j}| \tag{5}$$

Our approach focuses on using $\{\mathbf{z}_{mK}\}_{m=1}^M$ as anchor points and aims to transform their associated distributions $\{q_{mK}\}_{m=1}^M$ into effective proposal distributions for estimating the probabilities of the $M$ nested subset events $\{P[\Omega_{a_m}]\}_{m=1}^M$. Our key motivation is that we have the freedom to make the distinction between $\Omega_{a_m}$ and $\Omega_{a_{m+1}}$ to be small. Consequently, the shift from $q_{mK}$ to $q_{(m+1)K}$ is also expected to be marginal and to be easily learned by the NF model through $K$ function transformations $\{\mathbf{f}_{mK+i}\}_{i=1}^K$. In the following, we describe an $M$-step training process, where the $m$-th step aims to train $q_{mK}$.

### 3.1 STEP 1: TRAINING $q_K$ ASSOCIATED WITH $\Omega_{a_1}$

Let us for now ignore all components after $\mathbf{z}_K$ in Figure 1 and focus on training $\{\mathbf{f}_i\}_{i=1}^K$ to produce $q_K$ as an effective proposal distribution for estimating the probability $P[\Omega_{a_1}]$. As the data generating distribution $p$ in our concerned problem is easy to evaluate and sample from, we could take it as the NF's base distribution, i.e., $q_0 = p$.

To begin with, we modulate the data generating distribution $p$ to produce a distribution $p_1^\tau \in \mathcal{P}^D$:

$$p_1^\tau(\mathbf{x}) \propto \begin{cases} p(\mathbf{x}) \cdot e^{-\tau(g(\mathbf{x})-a_1)} & \text{when } g(\mathbf{x}) > a_1 \\ p(\mathbf{x}) & \text{when } g(\mathbf{x}) \le a_1 \end{cases} = \frac{1}{Z} e^{\min(\tau(a_1-g(\mathbf{x})),0)} p(\mathbf{x}) \quad (6)$$

where $\tau > 0$ is a temperature hyper-parameter, and $Z$ is a normalization constant ensuring valid distribution. Recall that the condition $g(\mathbf{x}) > a_1$ is equivalent to $\mathbf{x} \notin \Omega_{a_1}$, we can understand that $p_1^\tau$ essentially compresses the height of $p(\mathbf{x})$ when $\mathbf{x}$ lies outside the set $\Omega_{a_1}$, and the extent of this compression is determined by the margin between $g(\mathbf{x})$ and $a_1$. Next, we use $p_1^\tau$ as a target to learn a proposal distribution that allows for easy sampling. Noticing that any distribution defined in the NF model (such as the one we consider here, $q_K$) is easy to sample from, we minimize the following KL divergence loss to drive $q_K$ to be close to $p_1^\tau$:

$$D[q_K || p_1^\tau] = \int q_K(\mathbf{z}_K) \log \frac{q_K(\mathbf{z}_K)}{p_1^\tau(\mathbf{z}_K)} \, d\mathbf{z}_K \approx \frac{1}{N} \sum_{n=1}^{N} \log \frac{q_K(\mathbf{z}_K^n)}{p_1^\tau(\mathbf{z}_K^n)}, \quad \mathbf{z}_K^n \sim q_K(\cdot)$$

$$\approx \frac{1}{N} \sum_{n=1}^{N} \left[ \log p(\mathbf{z}_0^n) - \sum_{j=1}^{K} \log |\det \mathbf{J}_{\mathbf{f}_j}^n| - \log p_1^\tau(\mathbf{z}_K^n) \right], \quad \mathbf{z}_0^n \sim q_0(\cdot) \quad (7)$$

$$\propto -\frac{1}{N} \sum_{n=1}^{N} \sum_{j=1}^{K} \log |\det \mathbf{J}_{\mathbf{f}_j}^n| - \frac{1}{N} \sum_{n=1}^{N} \log p_1^\tau(\mathbf{f}_{K:1}(\mathbf{z}_0^n)), \quad \mathbf{z}_0^n \sim p(\cdot)$$

where in the second line, we do change of variables, and use Eq. (4) and $q_0 = p$. In the last line, we use the short notation $\mathbf{z}_K^n = \mathbf{f}_{K:1}(\mathbf{z}_0^n) = \mathbf{f}_K \circ \mathbf{f}_{K-1} \circ \cdots \circ \mathbf{f}_1(\mathbf{z}_0^n)$ and omit those terms don't depend on the learnable functions $\{\mathbf{f}_i\}_{i=1}^{K}$. Note that the normalization constant $Z$ in $p_1^\tau$ is not needed in the computation, as it will appear as a constant $\log Z$ in Eq. (7) which won't affect training.

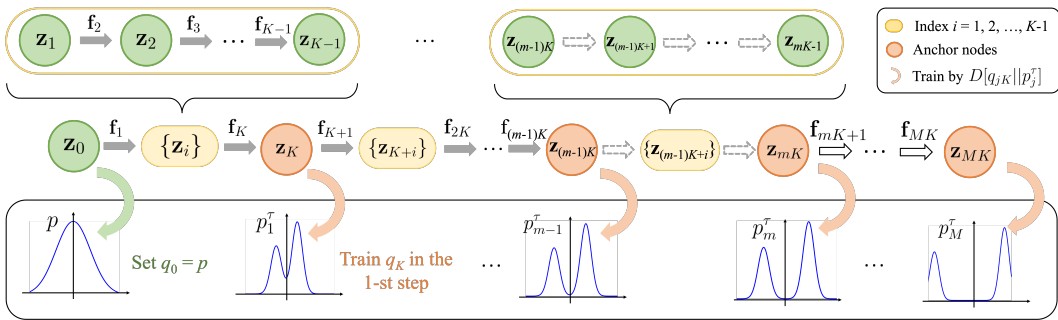

Figure 1: An illustration of our proposed NOFIS approach. Nodes $\{\mathbf{z}_{jK}\}_{j=1}^{M}$ along the normalizing flow highlighted in orange serve as anchor points. The distributions $\{q_{jK}\}_{j=1}^{M}$ associated with these nodes will be learned to align with the constructed target distributions $\{p_j^\tau\}_{j=1}^{M}$, achieved by adjusting the functions $\{\mathbf{f}_i\}_{i=1}^{MK}$. When learning $q_{mK}$, the gray-filled arrows represent frozen functions, the gray dashed-line arrows are learnable, while the gray solid-line arrows are yet to be trained.

**Important Remarks.** Several important clarifications must be made. Firstly, the NF model utilizes specific network architectures to parameterize $\mathbf{f}_i(\cdot)$ as $\mathbf{f}(\cdot; \boldsymbol{\theta}_i)$. It is crucial to meticulously design the form of $\mathbf{f}(\cdot; \boldsymbol{\theta}_i)$ (Dinh et al., 2014; 2016), to ensure that the evaluation of the determinant of its Jacobian matrix, as required by Eq. (7), is straightforward. Secondly, we have the option to employ the learned $q_K$ for estimating $P[\Omega_{a_1}]$ by incorporating it with the IS approach. However, we won't pursue it as our sole objective is the final rare event probability $P[\Omega_{a_M}] = P[\Omega]$. Namely, learning $q_K$ is for ease of learning subsequent distributions such as $q_{2K}$, $q_{3K}$, and ultimately $q_{MK}$. Thirdly, it is advisable to select the hyper-parameter $a_1$ in such a way that $P[\Omega_{a_1}]$ is not too small (e.g., greater than 0.1). Because it ensures an adequate number of samples $\mathbf{z}_K^n$ are located within $\Omega_{a_1}$, which makes the training perform effectively. This is indeed achievable, because when $a_1 \to \infty$, $P[\Omega_{a_1}] \to 1.0$. Alternatively, it should be noted that this also explains why training a proposal distribution directly associated with $\Omega_{a_M}$ is not feasible, as $P[\Omega_{a_M}]$ is extremely small and obtaining samples within $\Omega_{a_M}$ becomes nearly impossible.

Fourthly, based on Eq. (3), we know that the theoretically optimal proposal distribution for estimating $P[\Omega_{a_1}]$ is proportional to $p(\mathbf{x})\mathbb{1}[\mathbf{x} \in \Omega_{a_1}]/P[\Omega_{a_1}]$. For convenience, we denote this best proposal as $p_1^{\infty}$ for the reason that it is the limit of $p_1^{\tau}$ when $\tau \to \infty$. It seems appealing to use $p_1^{\infty}$ as the target in Eq. (7) instead of $p_1^{\tau}$. However, we observe that it brings severe training issues. To illustrate, if there exists a sample $\mathbf{z}_K^n = \mathbf{f}_{1:K}(\mathbf{z}_0^n)$ located outside $\Omega_{a_1}$, then $p_1^{\infty}(\mathbf{f}_{1:K}(\mathbf{z}_0^n))$ strictly equals zero, rendering the training loss undefined. On the other hand, if all sampled $\mathbf{z}_K^n$'s locate inside $\Omega_{a_1}$, then we actually drive $q_K$ to the data generating distribution $p$ because $p_1^{\infty}(\mathbf{f}_{1:K}(\mathbf{z}_0^n)) \propto p(\mathbf{f}_{1:K}(\mathbf{z}_0^n))$ holds true for all $n$ and the normalization constant doesn't matter when training with Eq. (7). Refer to Appendix A for more details on the temperature hyper-parameter.

Finally, Eq. (7) is usually referred to as the *reverse* KL divergence (Bishop & Nasrabadi, 2006). Alternatively, when swapping the places of $p_1^{\tau}$ and $q_K$, the *forward* KL divergence $D[p_1^{\tau}||q_K]$ could still measure the distribution difference. Consequently, one might consider using the forward KL divergence as an alternative training objective to replace Eq. (7). However, when we experiment with this forward KL divergence loss, we discover that a reweighting trick is needed and it performs significantly worse than the reverse KL loss. More detailed discussions are deferred to Appendix B.

## 3.2 Step $2 \sim M$: Training $q_{mK}$ by Freezing $q_{(m-1)K}$

Once the successful learning of $q_K$ is achieved through the training of $\{\mathbf{f}_i\}_{i=1}^K$ using the approach discussed in the previous subsection, we could train $\{\mathbf{f}_{K+i}\}_{i=1}^K$ to learn a subsequent $q_{2K}$ working as a proposal distribution for $\Omega_{a_2}$ similarly by minimizing $D[q_{2K}||p_2^{\tau}]$. To facilitate our discussion, we will describe a general $m$-th step, where $m$ is any integer between 2 and $M$. At the beginning of the $m$-th step, all functions $\{\mathbf{f}_i\}_{i=1}^{(m-1)K}$ are trained such that $q_{jK}$ is an effective proposal distribution associated with $\Omega_{a_j}$, for any $j = 1, 2, \cdots, m - 1$. Our goal in this step is to train $\{\mathbf{f}_{(m-1)K+i}\}_{i=1}^K$ to enforce $q_{mK}$ working as an effective proposal distribution for $\Omega_{a_m}$. Similar to Eq. (6) and (7), we use the following training loss:

$$D[q_{mK}||p_m^{\tau}] \propto -\frac{1}{N}\sum_{n=1}^{N}\sum_{j=1}^{mK}\log|\det \mathbf{J}_{\mathbf{f}_j}^n| - \frac{1}{N}\sum_{n=1}^{N}\log p_m^{\tau}(\mathbf{f}_{mK:1}(\mathbf{z}_0^n)), \quad \mathbf{z}_0^n \sim p(\cdot) \quad (8)$$

where $p_m^{\tau} \in \mathcal{P}^D$ is a constructed target distribution:

$$p_m^{\tau}(\mathbf{x}) = \frac{1}{Z}e^{\min(\tau(a_m - g(\mathbf{x})), 0)} p(\mathbf{x}) \quad (9)$$

**Freezing the Learned.** When minimizing Eq. (8), the functions $\{\mathbf{f}_i\}_{i=1}^{(m-1)K}$ will be held constant (as indicated by the gray-filled arrows in Figure 1). Our focus will solely be on training the functions $\{\mathbf{f}_{(m-1)K+i}\}_{i=1}^K$, which are represented by the gray dashed-line arrows in Figure 1. Recall that $q_{mK}$ is related to $q_{(m-1)K}$ through the learnable transformations $\{\mathbf{f}_{(m-1)K+i}\}_{i=1}^K$ and that the distribution $q_{(m-1)K}$ has already been well calibrated matching to $\Omega_{a_{m-1}}$. Consequently, there is no compelling reason to further train the previous $\mathbf{f}_i$'s (where $i \le (m-1)K$) in the $m$-th step, as $\{\mathbf{f}_{(m-1)K+i}\}_{i=1}^K$ alone possess ample expressive power to capture the distribution shift from $p_{m-1}^{\tau}$ to $p_m^{\tau}$ effectively. An alternative view is that we progressively expand the NF in each step by fixing the already learned transformations, and subsequently appending and training $K$ new transformations at the right end of the NF. We emphasize that this step-by-step training approach provides an implicit initialization method and enables feasible training. Namely, in the $m$-th step, $q_{(m-1)K}$ has already been learned to match $p_{m-1}^{\tau}$ which concentrates most of its mass inside $\Omega_{a_{m-1}}$, and thus, the sampled $\mathbf{z}_{(m-1)K}^n$ will have a high probability of lying within it. Given the default initialization where $\{\mathbf{f}_{(m-1)K+i}\}_{i=1}^K$ are close to identity functions, it follows that $\mathbf{z}_{mK}^n \approx \mathbf{z}_{(m-1)K}^n$ in the first epoch of the $m$-th step. When $\Omega_{a_m}$ doesn't change drastically compared to $\Omega_{a_{m-1}}$, a sufficient number of samples $\mathbf{z}_{mK}^n$ will lie within $\Omega_{a_m}$. This is crucial for the training process in the $m$-th step to advance effectively.

## 3.3 Summary and Implementation Details

Algorithm 1 summarizes the major steps of the proposed NOFIS approach for rare event probability estimation. It is worth mentioning that the NOFIS method necessitates a total of $(MEN + N_{\text{IS}})$ function calls to $g(\cdot)$. We empirically find that NOFIS is suitable to estimate $P_r \le 10^{-4}$, otherwise, the advantages of NOFIS over MC may be limited given the same function call budget. We will provide a quantitative explanation of this observation in the numerical result section.

**Choosing Hyper-parameters.** Firstly, to estimate probabilities $P_r \approx 10^{-x}$ (where $x$ is a positive integer), we empirically find that choosing $M$ equals $x$ is adequate. This observation aligns with previous experiences (Au & Beck, 2001; Sun & Li, 2014). As a rule of thumb, $\{a_m\}_{m=1}^M$ should approximately make the elements in $\{P[\Omega_{a_m}]\}_{m=1}^M$ scaled by 0.1 in order. Secondly, regarding the temperature hyper-parameter $\tau$, let us consider two points $\mathbf{x} \in \Omega_{a_m}$ and $\mathbf{x}' \notin \Omega_{a_m}$. Then our constructed $p_m^\tau$ should satisfy the constraint: $p_m^\tau(\mathbf{x}) \geq p_m^\tau(\mathbf{x}')$ for it to be meaningful as a target. Substituting the expression of $p_m^\tau$ as shown in Eq. (9) into this inequality results in a lower bound on $\tau$. Moreover, as we discussed in the fourth remark in Section 3.1, $\tau$ cannot be excessively large either. For more details, please refer to the ablation studies in Section 4.2 and Appendix A.

**Necessity of Learning.** If our sole objective is to estimate $P[\Omega_{a_1}]$ which is around 0.1, we don't need learning at all. Instead, we could do MCMC sampling from $p_1^\tau$ combined with IS estimation, or even perform MC sampling from $p$. However, neither of these two approaches could be directly adapted to estimate $P_r = P[\Omega_{a_M}]$. For example, MC would likely yield a trivial estimate of $P_r = 0$ because all generated samples lie outside $\Omega_{a_M}$. At this point, a natural thought is to utilize the nested subset events $\{\Omega_{a_m}\}_{m=1}^M$ to simplify the task. Because estimating $\{P[\Omega_{a_m}]\}_{m=1}^M$ in a sequential manner could be potentially easier than directly estimating $P[\Omega_{a_M}]$. Essentially, our NOFIS approach implements this thought, with the key being the memorization of $\Omega_{a_{m-1}}$ and its associated $p_{m-1}^\tau$ through $q_{(m-1)K}$ in the NF. This enables the subsequent learning of

---

**Algorithm 1** NOFIS

1: Provide a data generating distribution $p \in \mathcal{P}^D$ and an integral region $\Omega = \{\mathbf{x} \in \mathbb{R}^D | g(\mathbf{x}) \leq 0\}$.
2: Define a NF characterized by a base distribution $q_0 = p$, and a series of invertible transformations $\{\mathbf{f}_i(\cdot) = \mathbf{f}(\cdot; \boldsymbol{\theta}_i) : \mathbb{R}^D \to \mathbb{R}^D\}_{i=1}^{MK}$.
3: Choose hyper-parameters: (i) a strictly decreasing sequence $\{a_m\}_{m=1}^M$ satisfying $a_M = 0$, and (ii) the temperature hyper-parameter $\tau > 0$.
4: **for** $m = 1$ to $M$ **do**
5:      If $m \geq 2$, freeze $\{\boldsymbol{\theta}_i\}_{i=1}^{(m-1)K}$.
6:      **for** $e = 1$ to $E$ **do**
7:          Draw $N$ samples $\{\mathbf{z}_0^n\}_{n=1}^N$ from the base $q_0$.
8:          Calculate the loss $D[q_{mK}||p_m^\tau]$ using Eq. (8).
9:          Perform backward propagation and update the model parameters $\{\boldsymbol{\theta}_{(m-1)K+i}\}_{i=1}^K$.
10:      **end for**
11: **end for**
12: Return $P_r^{\text{IS}}$ using the learned $q_{MK}$ as the proposal distribution based on Eq. (2).

---

$\Omega_{a_m}$ to become manageable, because $\Omega_{a_m}$ is chosen to only have minor change from $\Omega_{a_{m-1}}$, and sampling from $q_{(m-1)K}$ is analytically tractable due to the NF model.

**Variants of Implementations.** We re-iterate that our approach, as outlined in Algorithm 1, follows a step-by-step training procedure. In contrast, various implementation variations exist. Firstly, by eliminating the external iteration on $m$ (i.e., setting $m = M$) and updating all $\{\boldsymbol{\theta}_i\}_{i=1}^{MK}$ in Step 9 (i.e., without freezing), we arrive at a variant that directly minimizes $D[q_{MK}||p_M^\tau]$ to learn all transformations. Building upon the modifications from the initial variant, we could employ $1/M \sum_{m=1}^M D[q_{mK}||p_m^\tau]$ as the loss, yielding the second variant. Nevertheless, we find that neither of these variants functions properly. Using $D[q_{MK}||p_M^\tau]$ as losses merely disregards all anchors in the middle, making it challenging to train the NF. As for the second variant, it raises questions about the validity of aggregating all $D[q_{mK}||p_m^\tau]$ values using their mean.

Lastly, solely eliminating Step 5 from Algorithm 1 leads to a version without freezing. As will be demonstrated in our ablation studies, this unfrozen variant does not exhibit superiority over our current frozen version, but it is evident that the unfrozen approach demands more computational resources. As a result, we have opted for the present step-by-step training procedure with freezing.

## 4 NUMERICAL RESULTS

As discussed at the beginning of Section 2, we set the data generating distribution $p$ as a standard Gaussian distribution $\mathcal{N}(\mathbf{0}, \mathbf{I})$ for all of our numerical experiments. Unless explicitly stated, we utilize RealNVP (Dinh et al., 2016) as the backbone NF model. In the subsequent first subsection, we present visualizations of several 2D test cases, assuming an unlimited number of function calls to $g(\cdot)$. Its primary objective is to qualitatively justify that our NOFIS approach can learn a $q_{MK}$ fully recovering the optimal proposal distribution, in an ideal scenario where there is *no limit on function calls*. Conversely, *the limited function call* scenario represents the practical situation when

deploying the algorithm. We quantitatively evaluate NOFIS's performance in the second subsection under this restricted scenario, followed by a few ablation studies in the end.

### 4.1 QUALITATIVE ANALYSIS

Figure 2 shows the learned $q_{MK}$ in various 2D cases; detailed settings are provided in Appendix C. Taking Figure 2 (b) as an example, we consider the integral region $\Omega = \{(x_1, x_2) \mid g(x_1, x_2) \leq 0\}$, where $g(x_1, x_2) = \min[(x_1+3.8)^2 + (x_2+3.8)^2, (x_1-3.8)^2 + (x_2-3.8)^2] - 1$. The best proposal distribution $q^\star$ defined in Eq. (3) is shown in the top row of Figure 2 (b). It is evident that $q^\star$ lies at the tail of the original data generating distribution $p$. Directly using an NF model to learn this $q^\star$ is not feasible due to numerical issues in training.

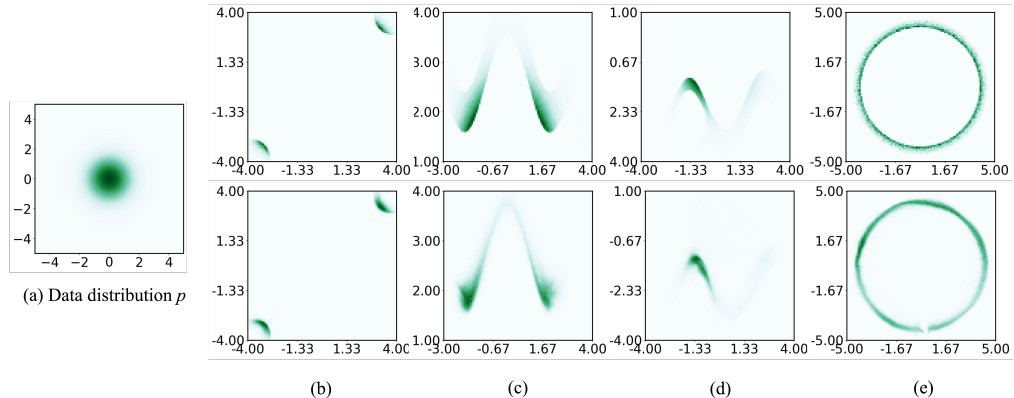

Figure 2: (a) The heatmap represents the data generating distribution $p = \mathcal{N}(\mathbf{0}, \mathbf{I})$. (b)-(e) The top row displays the theoretically optimal proposal distribution $q^\star$ defined in Eq. (3), while the bottom row illustrates the learned proposal distribution $q_{MK}$ generated by the NF using Algorithm 1. They exhibit a strong alignment in every case. When we overlay the highlighted green areas in (b)-(e) onto (a), we notice these areas occur at the tail of distribution $p$.

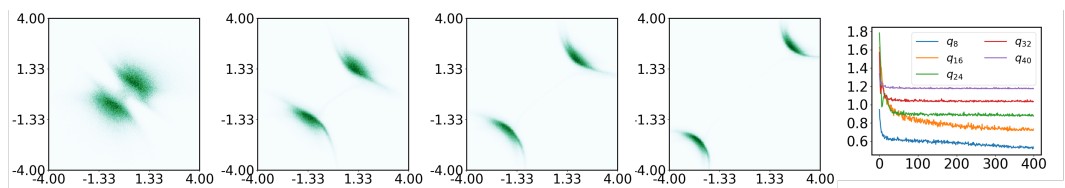

Figure 3: (a)-(d) The intermediate distributions $\{q_8, q_{16}, q_{24}, q_{32}\}$ of the NF model are plotted. They have been successfully trained, and the highlighted regions are centered at $(\pm 3.8, \pm 3.8)$ with radii that match our expected expression $\sqrt{a_m + 1}$. (e) The training loss in each step is plotted against the epoch. For better visualization, the Y-axis is presented on a logarithmic scale.

We set $K = 8$ and $M = 5$ in our NOFIS approach, so $\{q_8, q_{16}, q_{24}, q_{32}, q_{40}\}$ will be taken as anchors matching to $\{p_1^\tau, p_2^\tau, p_3^\tau, p_4^\tau, p_5^\tau\}$. To further justify our approach, we visualize intermediate distributions $\{q_8, q_{16}, q_{24}, q_{32}\}$ in Figure 3 (a)-(d), while $q_{40}$ is already displayed in the bottom row of Figure 2 (b). The region $\Omega_{a_m}$ induced by $a_m$ encompasses two circles centered at $(\pm 3.8, \pm 3.8)$ with a radius of $\sqrt{a_m + 1}$. According to Eq. (3), the heatmap of the optimal proposal distribution for estimating $P[\Omega_{a_m}]$ corresponds to "modulating/coloring" $\Omega_{a_m}$ based on the magnitude of $p$, resulting two thin leaf shape as exemplified in the top row of Figure 2 (b). Furthermore, as $a_m$ decreases alongside $m$, the radius also decreases, leading to a gradual outward shift of the two thin leaves from the origin. This phenomenon could indeed be observed in Figure 3 (a)-(d). Moreover, $\{a_1, a_2, a_3, a_4, a_5\}$ are set to $\{26, 15, 8, 3, 0\}$ in this case, and the radii of the learnt leaf shapes in Figure 2 (a)-(d) are surely consistent with the expression $\sqrt{a_m + 1}$. Last but not least, training loss curves are plotted in Figure 3 (e).

## 4.2 QUANTITATIVE SYNTHETIC AND REAL-WORLD EXPERIMENTS

We have shown the learned $q_{MK}$ could recover the optimal proposal distribution $q^\star$ provided an unlimited number of function calls. However, our primary objective is not to achieve this level of accuracy. Instead, our focus is on estimating the small probability, for which a learned $q_{MK}$ relatively close to $q^\star$ will be adequate. In this subsection, we will demonstrate that only a few function calls are necessary for this purpose, making the proposed NOFIS approach comparable to or even superior to baseline methods. Specifically, We take into account six methods as our baseline. The evaluation of algorithm performance is based on two metrics: (i) the number of function calls and (ii) the prediction error measured in the logarithm. **For complete reproducibility**, readers can find detailed experiment setups and algorithm settings in Appendix C.

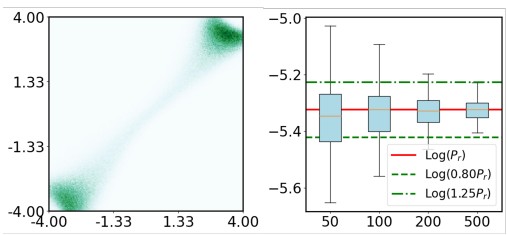

Figure 4: Left: The learned $q_{MK}$ for Case (#1) in a single run with 32K function calls. Right: Utilize this acquired $q_{MK}$ to generate an IS estimator with varying $N_{IS}$. The X-axis and Y-axis denote $N_{IS}$ and logarithm probability, respectively.

Table 1 presents the rare event estimation outcomes using 5 benchmark functions. Taking the case (#1) Leaf as an example, our NF model is trained using $M = 4$ steps, $E = 20$ epochs, and a batch size of $N = 400$, resulting in a total of $MEN = 32000$ function calls. Additionally, generating the IS estimator requires extra $N_{IS} = 20$ function calls in the end. The left part of Figure 4 showcases the learned proposal distribution $q_{MK}$, and the right part further reveals that when increasing $N_{IS}$, the estimation could become even more accurate. It is worth noting that the Leaf test case here is precisely the one depicted in Figure 2 (b). Comparing the left part of Figure 4 to the lower part of Figure 2 (b), we conclude limiting the number of function calls leads to a degradation in the learned proposal distribution, but NOFIS still successfully captures the two-leaf shape and generates highly accurate probability estimates.

As shown in Table 1, NOFIS consistently attains the lowest error while requiring the fewest function calls across all test cases, outperforming the other baseline methods. Notably, we observe that Adapt-IS exhibits inferior performance in high-dimensional test cases, which aligns with findings in (Biondini, 2015). Furthermore, SSS might be ineffective in test cases where the volume of $\Omega$ is small because it relies on scaling up the standard deviation (Sun & Li, 2014).

Table 1: Results on synthetic experiments averaged from 20 runs are reported in the format 'number of calls / logarithm error'. Here 'K' represents one thousand, and '—' indicates algorithm failure.

|  | (#1) Leaf | (#2) Cube | (#3) Rosen | (#4) Levy | (#5) Powell |
|---|---|---|---|---|---|
| Dimension | 2 | 6 | 10 | 20 | 40 |
| Golden $P_r$ | 4.74E-6 | 2.15E-9 | 4.69E-4 | 3.70E-6 | 3.15E-05 |
| MC | 50.0K / 9.11 | 500K / 11.33 | 7.0K / 1.87 | 50.0K / 11.80 | 10.0K / 11.0 |
| SIR | 50.0K / 9.30 | 500K / 10.62 | 7.0K / 0.96 | 50.0K / 14.56 | 10.0K / 3.66 |
| SUC | 47.5K / 4.79 | 279.9K / 7.28 | 8.3K / 0.85 | 50.0K / 4.31 | 9.6K / 3.52 |
| SUS | 42.0K / 0.23 | 206.0K / 0.096 | 7.0K / 0.40 | 49.0K / 0.53 | 9.0K / 5.80 |
| SSS | 40.0K / 0.70 | 400.0K / 1.53 | 8.0K / 0.46 | — | 8.0K / 0.84 |
| Adapt-IS | 35.0K / 0.25 | 227.0K / 6.23 | 8.4K / 15.07 | 56.0K / 9.20 | 7.9K / 15.56 |
| NOFIS (ours) | 32.0K / 0.11 | 197.5K / 0.078 | 7.0K / 0.32 | 48.2K / 0.44 | 7.0K / 0.38 |

Table 2 displays the outcomes of rare event estimation obtained from five real-world experiments spanning diverse domains. Each of these test cases revolves around the probability that a system's performance degradation (e.g., the Gain of the Opamp in (#1)) surpasses a specific threshold due to variations in system parameters (e.g., the width/length of CMOS transistors in (#1) Opamp). Further details about each case can be found in Appendix C. NOFIS has demonstrated superior performance in real-world test cases, achieving the smallest error with the fewest function calls in most scenarios, except for the last ResNet case where it performed slightly worse than SUS.

Table 2: Results from real-world experiments, averaged from 20 runs, are reported in the following format: 'number of calls / logarithm error', except in the case (#5), which is repeated four times.

| | (#1) Opamp | (#2) Oscillator | (#3) CP | (#4) Y-branch | (#5) ResNet |
|---|---|---|---|---|---|
| Dimension | 5 | 6 | 16 | 26 | 62 |
| Golden $P_r$ | 1.30E-5 | 1.81E-6 | 5.75E-6 | 4.27E-5 | 6.00E-5 |
| MC | 100K / 5.4 | 100K / 13.58 | 100K / 8.27 | 50K / 2.52 | 20K / 4.16 |
| SIR | 50K / 3.63 | 50K / 0.24 | 100K / 8.73 | 50K / 4.18 | 20K / 8.13 |
| SUC | 49K / 3.58 | 40.1K / 4.33 | 50.5K / 3.66 | 23.9K / 2.84 | 22.9K / 3.62 |
| SUS | 45K / 0.08 | 45K / 0.13 | 45K / 0.15 | 35.0K / 0.18 | 20K / 0.55 |
| SSS | 60K / 0.85 | 40K / 1.17 | 40K / 1.31 | 40K / 0.30 | 20K / 3.12 |
| Adapt-IS | 48K / 2.89 | 43K / 2.62 | 43K / 12.77 | 43K / 15.28 | — |
| NOFIS (ours) | 45K / 0.07 | 31K / 0.12 | 35K / 0.12 | 32.5K / 0.11 | 18K / 0.61 |

**Ablation Studies.** We examine the effects of various implementation choices on the performance of NOFIS using Opamp, CP, and Y-branch. The results presented in Table 2 are labeled as the "nominal" configuration. The left segment of Figure 5 displays the prediction error when a single incremental change is applied to the nominal setup. For the 'LongThre' parameter, we set $M = 9$, and for 'SmallTemp', we use $\tau = 1$, whereas the nominal settings have $M \in [4, 6]$ and $\tau \in [10, 30]$. It's noteworthy that altering the freezing approach, using extended threshold sequences, or employing smaller temperatures doesn't consistently lead to improvements in NOFIS performance.

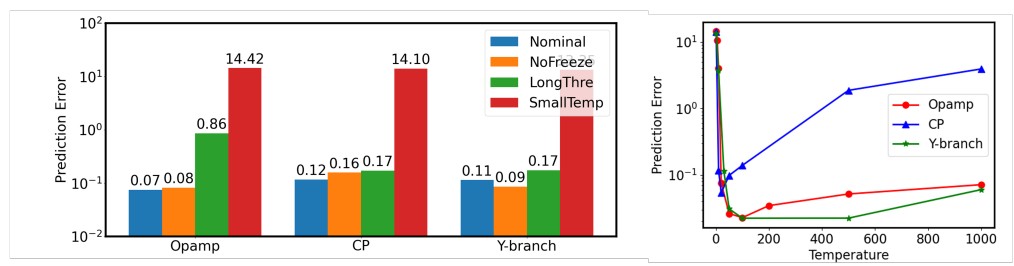

Figure 5: Left: Ablation studies are carried out on non-freezing, long threshold sequences (i.e., large $M$), and small temperature $\tau$. Right: The error of NOFIS is plotted versus the temperature $\tau$.

Moreover, the right part of Figure 5 uncovers two significant observations: (i) NOFIS demonstrates great robustness within the temperature range of $\tau \in [10, 200]$, and (ii) a carefully tuned temperature $\tau$ could potentially yield **even better outcomes** for the proposed NOFIS method. For example, the optimal results (depicted by the lowest markers) on the red Opamp, blue CP, and green Y-branch curves in the right section of Figure 5 achieve prediction errors of 0.026, 0.054, and 0.023, respectively. These estimation errors are considerably smaller than their counterparts (i.e., 0.07, 0.12, and 0.11) reported in Table 2, while utilizing the same number of function calls.

## 5  CONCLUSIONS AND LIMITATIONS

In this paper, we introduce NOFIS, an efficient method for estimating rare event probabilities through normalizing flows. NOFIS learns a sequence of functions to shift a base distribution towards an effective proposal distribution, using nested subset events as bridges. Our qualitative analysis underscores NOFIS's adeptness in accurately recovering the optimal proposal distribution. Our quantitative exploration across 10 test cases justifies NOFIS's superiority over six baseline methods.

The effectiveness of NOFIS hinges on accurately configuring nested subset events. Yet, the prevailing approach, both in this work and previous studies (Au & Beck, 2001; Sun & Li, 2014), entails human intervention. Developing an automated method for defining nested subset events stands as a crucial avenue for future research.

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

## A  THE TEMPERATURE HYPER-PARAMETER

Here we analyze the impact of the temperature hyper-parameter $\tau$. To begin with, we revisit the fourth remark in Section 3.1 and explain why $p_1^\infty$ is not a valid target distribution for training. According to Eq. (6), we know:

$$p_1^\infty(\mathbf{x}) = \begin{cases} p(\mathbf{x})/Z & \text{when } g(\mathbf{x}) > a_1 \\ 0 & \text{when } g(\mathbf{x}) \le a_1 \end{cases} \tag{10}$$

When employing Eq. (7) to train the NF with $p_1^\infty$ as the intended target, two scenarios may unfold: (i) at least one sample $\mathbf{z}_K^n$ lies beyond $\Omega_{a_1}$, and (ii) all $\mathbf{z}_K^n$ instances are situated within $\Omega_{a_1}$. In the first scenario, Eq. (10) indicates that the value of $p_1^\infty(\mathbf{z}_K^n)$ is rigorously zero, rendering Eq. (7) undefined. In the second scenario, Eq. (10) establishes that $p_1^\infty(\mathbf{z}_K^n) \propto p(\mathbf{z}_K^n)$ remains true for all $n$, thereby compelling Eq. (7) to essentially steer $q_K$ towards the original data distribution $p$. Both scenarios are problematic and deviate from our intended outcomes. Moreover, the distribution $p_1^\infty$ lacks continuity on the boundary of $\Omega_{a_1}$ (i.e., when $g(\mathbf{x}) = a_1$), but $p_1^\tau$ does not when $\tau \ne \infty$. Note that the aforementioned reasoning applies equally to any $p_m^\infty$.

The above also implies that opting for a large value of $\tau$ could potentially result in numerical instability during training (e.g., leading to small denominators in Eq. (7) and subsequently large KL loss values), thereby causing the performance of the proposed NOFIS method to deteriorate. Conversely, excessively small $\tau$ is also inadvisable. To justify, consider any pair of points $(\mathbf{x}, \mathbf{x}')$, where $\mathbf{x} \in \Omega_{a_m}$ and $\mathbf{x}' \notin \Omega_{a_m}$. For our constructed $p_m^\tau$ to retain its validity as a target, it should uphold the constraint: $p_m^\tau(\mathbf{x}) \ge p_m^\tau(\mathbf{x}')$. Employing Eq. (9), this constraint is converted to:

$$\tau \ge \frac{1}{g(\mathbf{x}') - a_m} \ln \frac{p(\mathbf{x}')}{p(\mathbf{x})} \tag{11}$$

which imposes a lower bound on $\tau$. For instance, when $p = \mathcal{N}(\mathbf{0}, \mathbf{I})$, the above inequality becomes:

$$\tau \ge \frac{1}{2} \frac{||\mathbf{x}||_2 - ||\mathbf{x}'||_2}{g(\mathbf{x}') - a_m} \tag{12}$$

Combining these insights, we anticipate that the plot illustrating NOFIS estimation error against temperature $\tau$ will assume a bowl-like shape, showcasing higher errors at both extremes when $\tau$ is excessively small or large. Indeed, this trend is observed in the right section of Figure 5.

## B  REVERSE AND FORWARD KL DIVERGENCE

Here we elucidate the last remark in Section 3.1 about reverse and forward KL divergence. By interchanging the positions of $p_1^\tau$ and $q_K$ in Eq. (7), we derive the forward KL divergence:

$$D[p_1^\tau || q_K] = \int p_1^\tau(\mathbf{z}_K) \log \frac{p_1^\tau(\mathbf{z}_K)}{q_K(\mathbf{z}_K)} \, d\mathbf{z}_K \approx \frac{1}{N} \sum_{n=1}^N \log \frac{p_1^\tau(\mathbf{z}_K^n)}{q_K(\mathbf{z}_K^n)}, \quad \mathbf{z}_K^n \sim p_1^\tau(\cdot) \tag{13}$$

This forward KL divergence necessitates sampling from $p_1^\tau$ instead of $q_K$. However, the task of sampling from $p_1^\tau$ is intricate, while $q_K$ represents a distribution within the NF model that can be sampled precisely. One solution is to employ importance sampling once more:

$$\begin{aligned} D[p_1^\tau || q_K] &= \int p_1^\tau(\mathbf{z}_K) \log \frac{p_1^\tau(\mathbf{z}_K)}{q_K(\mathbf{z}_K)} \, d\mathbf{z}_K = \int q_K(\mathbf{y}) \frac{p_1^\tau(\mathbf{z}_K)}{q_K(\mathbf{z}_K)} \log \frac{p_1^\tau(\mathbf{z}_K)}{q_K(\mathbf{z}_K)} \, d\mathbf{z}_K \\ &\approx \frac{1}{N} \sum_{n=1}^N \frac{p_1^\tau(\mathbf{z}_K^n)}{q_K(\mathbf{z}_K^n)} \log \frac{p_1^\tau(\mathbf{z}_K^n)}{q_K(\mathbf{z}_K^n)}, \quad \mathbf{z}_K^n \sim q_K(\cdot) \end{aligned} \tag{14}$$

We experimentally find that the NF distributions are not trained properly when using this forward KL loss. To intuitively understand this, our desired optimal solution $q_K$ should equal $p_1^\tau$, resulting in a forward KL loss of $D[p_1^\tau || q_K]$ equal to zero. However, due to the re-weighting introduced in Eq. (14) and the finite sample size $N$ employed during training, an alternative trivial solution

emerges. Specifically, imagine a $q_K$ concentrates its mass predominantly in regions where $p_1^\tau$ approaches zero. In theory, this would cause the associated $D[p_1^\tau || q_K]$ to approach infinity. However, in practice, when we undertake training in accordance with the approach outlined in the second line of Eq. (14), we are limited to drawing a finite number of samples. Consequently, the empirical forward KL has a high likelihood of approximating zero, as $\epsilon \log \epsilon \approx 0$ when $\epsilon \approx 0$. Thus, training with the forward KL might be problematic as it could lead to convergence towards this trivial solution.

## C DETAILED EXPERIMENTAL SETUPS

**Evaluation Metrics.** Our assessment of algorithm performance rests upon two fundamental metrics: (i) the count of function calls and (ii) the logarithmic prediction error, denoted as $err = |\log P_r^{\text{est}} - \log P_r|$, where $P_r^{\text{est}}$ presents the estimated rare event probability, and $P_r$ represents the reference or golden rare event probability.

Several clarifications are pertinent to this error expression. Firstly, the adoption of logarithmic error arises from the vital need to comprehend the order of magnitude of the rare event probability. This logarithmic error metric can directly mirror this magnitude. For instance, when $err < 1.0$, it signifies that the estimated probability is within one order of magnitude of the true probability. Secondly, within the domain of rare event estimation, adopting a linear error measure could be misleading. For instance, consider a scenario where the golden probability is $10^{-5}$. In a linear scale, probability estimates like $10^{-7}$ and $10^{-20}$ would both yield $err \approx 10^{-5}$, suggesting comparable accuracy. However, it is evident that $10^{-7}$ constitutes a more precise estimation than $10^{-20}$. Thirdly, accounting for instances where $P_r^{\text{est}}$ could be zero, to ensure a valid error definition, we actually implement the expression $err = |\log \max(P_r^{\text{est}}, 10^{-20}) - \log P_r|$ in our experiments. This formulation essentially designates $10^{-20}$ as the smallest sensible unit of accuracy in the decimal representation.

In a related context, it is noteworthy that except for the Cube case in Table 1, where the golden $P_r$ can be analytically computed, the golden $P_r$ values in all other experiments are estimated from a substantial number of Monte Carlo (MC) samples. Given the constraints of current computational resources, running MC simulations with sample sizes of $10^8 \sim 10^{10}$ is already considerably time-intensive. This rationale underscores why our numerical experiments predominantly operate with $P_r > 10^{-7}$, as otherwise, the accuracy of our golden $P_r$ itself might be compromised. Looking ahead, as computational resources continue to advance, we aspire to extend the algorithm's validation to numerical examples involving even smaller $P_r$ values in the future.

**Baseline Methods.** We consider six baseline methods in our experiments.

- MC: The conventional Monte Carlo method (Bucklew & Bucklew, 2004). MC involves drawing $N$ i.i.d. samples from the distribution $p$ and estimating the rare event probability by calculating the ratio of samples that fall within $\Omega$.

- SIR: An abbreviation for simple regression. SIR draws $N$ samples (e.g., $N = 10^4$) from a mixture of distribution $p$ and a uniform distribution over $[-T, T]^D \in \mathbb{R}^D$, where $T$ is a hyper-parameter. The simulator $g(\cdot)$ is then used to evaluate the corresponding function values. Subsequently, a deep neural network is trained on this paired data to learn the mapping $g(\mathbf{x})$. Afterwards, $N_{\text{eval}}$ samples (e.g., $N_{\text{eval}} = 10^9$) are generated from the distribution $p$, and their function values are evaluated using the neural network. The rare event probability estimation involves calculating the ratio of how many of these $N_{\text{eval}}$ samples fall within $\Omega$.

- SUS: An abbreviation for subset simulation (Au & Beck, 2001; Sun & Li, 2014). In short, SUS first defines a series of nested subset events, and then decomposes the original rare event probability estimation into estimating several conditional probabilities. This is accomplished using Markov Chain Monte Carlo (MCMC) techniques.

- SUC: An abbreviation for subset classification. In short, the MCMC sampling in SUS is replaced with modern deep neural networks. Specifically, $M$ binary classifiers are utilized, where the $m$-th classifier corresponds to the $m$-th subset event $\Omega_{a_m}$. In the $m$-th step, $N$ samples are generated from $p$, and the $(m-1)$-th classifier is used to determine if a sample falls within $\Omega_{a_{m-1}}$. Subsequently, the simulator $g(\cdot)$ is called on those samples predicted to be within $\Omega_{a_{m-1}}$, and a neural network is trained to classify whether a sample lies within $\Omega_{a_m}$.

- SSS: An abbreviation for scaled-sigma sampling (Sun et al., 2015). In scenarios where the data generating distribution $p$ follows a Gaussian distribution, SSS involves the manipulation of its

standard deviation, either by scaling it up or down. The approach proceeds by estimating the probabilities associated with these scaled distributions using MC. Subsequently, it employs extrapolation techniques to estimate the original rare event probability based on an analytical model.

- Adapt-IS: An abbreviation for adaptive importance sampling (Bucklew & Bucklew, 2004). Adapt-IS employs a family of Gaussian mixtures as the proposal distribution and optimizes the parameters of the Gaussian components using a cross-entropy loss function.

For fair comparisons, the SUC, SUS, and our proposed NOFIS method utilize identical nested subset events. Additionally, we maintain a comparable count of function calls across various algorithms.

**Testcase Details.** For the sake of simplicity, in the main text, we have parameterized the integral region $\Omega$ using a characteristic function $g(\mathbf{x}) \leq 0$. However, in real-world applications, the function $g(\cdot)$ holds tangible physical significance (e.g., representing the performance of a system characterized by $\mathbf{x}$). This function may encompass not only an upper bound of 0, but also a lower bound, e.g., $\Omega = \{\mathbf{x} \in \mathbb{R}^D | l \leq g(\mathbf{x}) \leq u\}$. Nevertheless, we can still retain the original representation: $\Omega = \{\mathbf{x} \in \mathbb{R}^D | \max [g(\mathbf{x}) - u, l - g(\mathbf{x})] \leq 0\}$ and follow the approach stated in the main text. Alternatively, we could have more flexibility by introducing subset events from both below and above. Namely, in parallel with Eq. (6) from the main text, we could define a subset event $\Omega_{l_1,u_1} = \{\mathbf{x} \in \mathbb{R}^D | l_1 \leq g(\mathbf{x}) \leq u_1\}$, utilizing a threshold pair $(l_1, u_1)$, and formulate the target distribution as follows:

$$p_1^\tau(\mathbf{x}) = \frac{1}{Z} e^{\min \tau (u_1 - g(\mathbf{x}), g(\mathbf{x}) - l_1, 0)} p(\mathbf{x}) \tag{15}$$

It depends on the user's preference whether to adopt this form or reformulate according to Eq. (6). Similarly, this thought can be extended to scenarios where the function $g(\cdot)$ returns a vector value.

Across all our experiments, we adopt RealNVP (Dinh et al., 2016) as the underlying NF model, and set $K = 8$. This means that there will be 8 learnable transformations (a.k.a., affine coupling layers in RealNVP) between two anchor points in Figure 1. Specifically, the first half of input dimensions remain unaltered in the affine coupling layers with odd indices, while the later half remain unchanged in those with even indices. The scaling and translation functions are both implemented using a feedforward neural network, consisting of three hidden layers, each comprising 128 neurons. Without explicitly mentioning, the temperature hyper-parameter is usually set to 10.

In the following, we present comprehensive information regarding the experimental setup for all of our numerical experiments, and to align with the real physical meanings, we adopt the format $\Omega = \{\mathbf{x} \in \mathbb{R}^D | l \leq g(\mathbf{x}) \leq u\}$ to describe the integral region $\Omega$.

- Figure 2 (b): The characteristic function is $g(x_1, x_2) = \min[(x_1 + 3.8)^2 + (x_2 + 3.8)^2, (x_1 - 3.8)^2 + (x_2 - 3.8)^2] - 1$. The integral region is defined solely by an upper bound $u = 0$. We set the epoch count to $E = 400$, the batch size to $N = 1000$, $N_{\text{IS}} = 50$, and $M = 5$.

- Figure 2 (c): The characteristic function has been slightly adapted from the third test energy function as outlined in (Rezende & Mohamed, 2015). In short, we shift the energy function by a small amount. The integral region $\Omega$ is defined solely by an upper bound $u = 0.001$. we set the epoch count to $E = 400$, the batch size to $N = 1000$, $N_{\text{IS}} = 50$, and $M = 3$.

- Figure 2 (d): Similar to the above (c), but the shift amount is different. The integral region $\Omega$ is defined solely by an upper bound $u = -0.6$. We set the epoch count to $E = 400$, the batch size to $N = 1000$, $N_{\text{IS}} = 50$, and $M = 3$.

- Figure 2 (e): The characteristic function is $g(x_1, x_2) = x_1^2 + x_2^2$. The integral region $\Omega$ is defined by both a lower bound $l = 16$ and an upper bound $u = 20.25$. We set the epoch count to $E = 400$, the batch size to $N = 1000$, $N_{\text{IS}} = 50$, and $M = 5$.

- Table 1 (#1) Leaf: The characteristic function and the integral region are identical to those in Figure 2 (b). However, due to the limited function call constraint, we set the epoch count to $E = 20$, the batch size to $N = 400$, $N_{\text{IS}} = 50$, and $M = 4$ in this case.

- Table 1 (#2) Cube: The characteristic function is $g(\mathbf{x}) = \max (1.8 - \mathbf{x})$. The integral region $\Omega$ is defined solely by an upper bound $u = 0$. We set the epoch count to $E = 55$, the batch size to $N = 500$, $N_{\text{IS}} = 5000$, and $M = 7$.

- Table 1 (#3) Rosen: The characteristic function is a 10-dimensional Rosenbrock function (Rosenbrock, 1960). The integral region $\Omega$ is defined by both a lower bound $l = 3.48$ and an upper

bound $u = 3.52$. We set the epoch count to $E = 15$, the batch size to $N = 100$, $N_{\text{IS}} = 1000$, and $M = 4$.

- Table 1 (#4) Levy: The characteristic function is a 20-dimensional Levy function. The integral region $\Omega$ is defined by both a lower bound $l = 0$ and an upper bound $u = 6$. We set the epoch count to $E = 20$, the batch size to $N = 400$, $N_{\text{IS}} = 200$, and $M = 6$.

- Table 1 (#5) Powell: The characteristic function is a 40-dimensional Powell function. The integral region $\Omega$ is defined solely by an upper bound $u = 4$. We set the epoch count to $E = 15$, the batch size to $N = 100$, $N_{\text{IS}} = 1000$, and $M = 4$.

- Table 2 (#1) Opamp: The random variable $\mathbf{x}$ represents the variation of width/length of MOS transistors in an three-stage Opamp circuit (Lyu et al., 2018). The characteristic function $g(\cdot)$ measures the Opamp Gain. The integral region is defined by an upper bound $u = 71.8$ dB. We set the epoch count to $E = 20$, the batch size to $N = 400$, $N_{\text{IS}} = 5000$, and $M = 5$.

- Table 2 (#2) Oscillator: The random variable $\mathbf{x}$ represents the variation of parameters in the toy car (Song et al., 2021). The characteristic function $g(\cdot)$ measures the displacement of the toy car. The integral region is defined solely by a lower bound $l = 2.6$ m. We set the epoch count to $E = 20$, the batch size to $N = 300$, $N_{\text{IS}} = 1000$, and $M = 5$.

- Table 2 (#3) CP: The random variable represents the variation of width/length of MOS transistors in a Charge Pump (CP) circuit (Gao et al., 2019). The characteristic function $g(\cdot)$ represents the current mismatch at the output. The integral region $\Omega$ is defined solely by an lower bound $u = 370$ uA. We set the epoch count to $E = 20$, the batch size to $N = 300$, $N_{\text{IS}} = 5000$, and $M = 5$.

- Table 2 (#4) Y-branch: The random variable $\mathbf{x}$ represents the boundary deformation of a silicon photonic Y-branch (Zhang et al., 2020). The characteristic function $g(\cdot)$ represents the power transmission from the input port to the output port. The integral region is defined solely by an upper bound $u = 31.7\%$. We set the epoch count to $E = 15$, the batch size to $N = 300$, $N_{\text{IS}} = 10000$, and $M = 5$.

- Table 2 (#5) ResNet: The random variable $\mathbf{x}$ represents the variation of parameters in a ResNet18. Roughly, this Gaussian noise $\mathbf{x}$ is added in a layer-wise manner, as shown in the pseudo code:

```
for i, param in enumerate(model.parameters()):
        param.data = param.data * (1 + x[i] * noise_std)
```

This approach is adopted due to the substantial number of parameters in ResNet18, which amounts to millions. Treating each parameter individually would be computationally prohibitive. The characteristic function represents the ResNet18 performance degradation on the test dataset. The integral region is solely defined by a lower bound. We set the epoch count to $E = 20$, the batch size to $N = 100$, $N_{\text{IS}} = 10000$, and $M = 4$.

We would like to emphasize once again that the experiments depicted in Figure 2 (b)-(e) were conducted without imposing any constraints on the number of function calls. As a result, the epoch count and batch size used in these instances are larger compared to other testcases.

