# OpenReview forum: "Rare Event Probability Learning by Normalizing Flows"
_ICLR.cc/2024/Conference — ICLR 2024 Conference Withdrawn Submission_

### Official Review · Reviewer_U9Yx · 2023-10-23

**Soundness:** 2 fair
**Presentation:** 3 good
**Contribution:** 1 poor
**Rating:** 3
**Confidence:** 4

**Summary:**

The authors introduce rare event sampling via normalizing flows. For this they parameterize the rare event set via a function $g$ such that the rare event set is the set of points where $g \leq 0$. Then they introduce a sequence of decreasing sets $\Omega_{a_i}$ such that this goes to $\Omega$ for $i = M$. Now a normalizing flow is trained for approximate each set $\Omega_{a_i}$, which corresponds to a temperature schedule for the rare event probability measure.  The normalizing flows are trained each on their own using the reverse KL and then the weights up to flow $i-1$ are frozen for training the flow $i$. The approach is benchmarked against other rare event sampling methods such as SUS, SIR, .. on toy examples of varying information.

**Strengths:**

The paper does a good job at explaining its approach. The experimental results seem impressive and its design choices seem well-motivated via ablation studies. Furthermore, using a normalizing flows makes a lot of sense for this kind of task.

**Weaknesses:**

1) I am not convinced of the novelty of this approach. This paper mostly cites pre 2021 papers. Please clarify the relation to more modern approaches such as [1,2].

2) The flows are trained with the reverse KL. This comes with some caveats. First one assumes differentiability of the function $g$. Please comment on whether this is realistic. Furthermore, the reverse KL is known to be mode seeking. I think for most applications in the field of rare event sampling it is crucial to cover all the modes of a density. There has been some recent line of work for normalizing flows such as [3] to overcome this but this seems like a major limitation.

3) Similarly, the evaluation should also include some measure of the distance to the true measure and not only the estimated probability. As far as I understand the paper, this should be possible.

4) Please also cite relevant papers such as [4], who introduced a kind of log det schedule for covering multimodal distributions, which I think is related to way the different $\Omega_{a_i}$ are constructed.

5) This paper does not come with any code. Do the authors intend to make their code public? Appendix C does not suffice for reproducibility in my opinion.

6) The heuristic why MCMC wont cut it for this problem makes sense for vanilla MH. But if one takes gradient informed steps such as HMC or MALA, I am not sure why this rationale outlined in section 3.3 should hold true. What is the proposal for MCMC taken in the experiments?

[1] A Flow-Based Generative Model for Rare-Event Simulation, Gibson et al

[2] Conditioning Normalizing Flows for Rare Event Sampling, Falkner et al

[3] Flow Annealed Importance Sampling Bootstrap, Midgley et al.

[4] Deep Probabilistic Imaging: Uncertainty Quantification and Multi-modal Solution Characterization for Computational Imaging , Sun et al.

**Questions:**

See weaknesses. I think the paper follows a nice idea, has several benchmarks, but does a poor job at literature review. Also I think uploading the code is very important for reproducibility, since this paper is mostly applied.

---

### Official Review · Reviewer_E1PQ · 2023-10-25

**Soundness:** 3 good
**Presentation:** 3 good
**Contribution:** 2 fair
**Rating:** 3
**Confidence:** 3

**Summary:**

The authors apply a normalizing flow model approach to rare event probability estimation, defined where the probability is less than 1e-4. This is done by the normalizing flow model learning proposal distributions, then estimating rare event probability using importance sampling on the learned proposal distribution.

**Strengths:**

Paper is well presented, and using normalizing flows to assist with importance sampling (as compared to the other way around which has been done) is new.

**Weaknesses:**

Freezing seems to provide only a marginal advantage over non-freezing. The main advantage as the authors proposed is in the speed, but that's not particularly central to the paper as speed is measured by function calls and not wall clock time. If we remove step 5 from NOFIS then most of the method is not particularly distinguishable from standard normalizing flows.

In addition, if we're looking for just samples from the proposal distribution, what's the advantage of using NFs over other generative models? If there is a lack of distinguishing feature then the middle portion on NFs specifically might not be needed in lieu for a general generative model construction.

**Questions:**

Figure 2: Overlay highlighted green areas - not sure if I see the highlights?

What about just using the normalizing flow to directly estimate the likelihood of the rare event?

---

### Official Review · Reviewer_27ee · 2023-10-26

**Soundness:** 3 good
**Presentation:** 3 good
**Contribution:** 1 poor
**Rating:** 3
**Confidence:** 4

**Summary:**

The paper introduces a technique for rare event sampling that combines normalizing flows with importance sampling. The authors refer to this technique as NOFIS (NOrmalizing Flows assisted Importance Sampling). They justify their work by highlighting the limitations of standard sampling algorithms, such as MCMC, in sampling regions of low probability, where the density, denoted as $p$, is approximately $10^{-X}$, with X being an integer greater than 4. In this context, known as the regime of rare event sampling, algorithms like MCMC would require an impractical number of samples, rendering these approaches highly inefficient. The authors propose that employing normalizing flows-aided importance sampling holds promise as a solution to this problem.

**Strengths:**

- The paper flows smoothly and is enjoyable to read.
- The authors provide great level of details and do not take anything for granted, which I appreciate.

**Weaknesses:**

- **Novelty**: I don’t find much novelty in the proposed paper. The technique presented by the authors has already been explored in many prior works in different fields, particularly in physics, where rare event sampling is often a challenging problem (see below).

- **Related Works**: Despite many prior works combining normalizing flows with importance sampling, and beyond, exist, this paper lacks a dedicated *Related Work* section. Several seminal works have been completely overlooked despite their significant contributions to the field of normalizing flow-aided importance sampling in statistical physics [1], chemistry[2], and quantum field theory[3,4,5].

- **Annealed Importance Sampling**: There is no reference to *annealed importance sampling* [6], which I believe is highly tight to the idea of the paper. Besides [6], several relevant works [7,8,9] perform annealed importance sampling within the context of normalizing flows, falling within the same category as the CRAFT method referenced in the paper, though only marginally. What these methods do closely aligns with what the authors propose in the paper: instead of learning the target distribution in one step, they 'anneal' towards that distribution by learning and sampling from intermediate distributions, ensuring that the final learned probability density has as much support as possible, including regions where the target density is small enough to fall within the rare event regime. I believe it is crucial for this paper to be published in this or any other venue to highlight the connection to these (and the previously referenced works).

- **Rare Event Sampling**: A recent paper [10] discusses similar behaviors in training normalizing flows and combining them with importance sampling to ensure full support over the target density, including rare event regions. I would find it interesting if the authors commented on this work within the context of their findings. Some of the metrics and tools proposed in [10], such as the mode-dropping estimator, could also be used to assess the performance of a sampler in approximating regions of low probability where a shallow sampler is likely to lose some of the probability mass.

- **Idea of Anchor Points**: The notion of *anchor points* has implicitly been explored in some of the prior works mentioned above, albeit with a slightly different connotation that may have escaped the authors' attention. For instance, in the paper by Kanwar et al. [4] (Fig. 4), the authors use a technique very similar to what is suggested in this paper, although with slightly different connotations (e.g., they use previously trained flow-based models as starting (anchor) points to sequentially train more challenging distributions).

- **Additional Related Works**: Other closely related works, such as [11], are not mentioned in the manuscript despite having similar titles. This may cause confusion for potential readers.

- **Experiments**: I find the results presented in the paper not entirely convincing. Although the authors compared their approach to a large set of baselines, this alone does not seem sufficient to claim the superiority of the proposed method. I am surprised that the proposed approach is not compared against prior works, such as Annealed Importance Sampling with Normalizing Flows [7], and naive RealNVP training with a sufficiently large number of couplings and no anchor points.

As a side note, I strongly recommend that the authors conduct an extensive literature search to include and acknowledge existing prior works, and eventually, compare and discuss potential differences and similarities

**Questions:**

- I'd like to see how the author would compare their work (and its corresponding novelty) to previous works. In particular, I'd like to see comparisons with Refs. [6-9] for the annealing aspect and Ref. [10] for the theoretical discussion regarding low-support regions (e.g., the rare event regime). Furthermore, discussing the differences concerning Ref. [11] would be helpful for the readers.

- I'd appreciate if the authors could perform an extensive literature search and create a Related Work section to place their paper in the context of existing prior works. Please refer to Refs. [1-11].

- I found the last paragraph in Section 3.1 and the discussion in Appendix B to be a bit unintuitive. It has been shown in the literature that using Forward KL, instead of Reverse KL, generally results in larger support and, therefore, has some benefits when combined with importance sampling. In that sense, I am surprised by the author's claim that training using Forward KL deteriorates performance. Do the authors consider the case where NO samples are given from the target density? If so, then I may understand this point. Otherwise, when a sample set from the target density, even if small, is available, it should be possible to show that training with Forward KL is feasible.

- It would be informative to see the density plot from Figure 4 for the other baselines as well.

- On page 8, referring to Figure 4, the authors write "[…] the right part further reveals that when increasing $N_{IS}$, the estimation could become even more accurate." This result does not seem neither novel nor unexpected. Indeed, it was already demonstrated in prior works, as seen in [1,5], that the variance of the importance sampling estimators scales with $N^{-1}$, with N being the number of samples. Could maybe the authors comment on this?

**Minor**

- The quality of the plots on pages 7-8 is quite poor. The axis labels are missing, and the font size for the x-y tick labels is too small.

- As a side note, I sometimes find the MK notation a bit confusing. However, I understand that it would require a substantial effort to rewrite the manuscript and adapt to a clearer notation. Nevertheless, this my be a feedback worth keeping in mind for the authors for future iterations of the manuscript.

- I find it somewhat unintuitive to completely relegate the discussion of the datasets to the appendix. Perhaps the authors could add corresponding references in the main text when mentioning the datasets and also refer to the Appendix for further details.

- In the conclusion, statements like *using nested subset events as bridges* agains strongly reminds of annealed importance sampling. I believe that a discussion comparing the present method to AIS, highlighting potential differences, or connecting them through their analogies is an essential element currently missing in the manuscript.


**References:**


- [1] [Nicoli, Kim A., et al. "Asymptotically unbiased estimation of physical observables with neural samplers." Physical Review E 101.2 (2020): 023304.](https://link.aps.org/accepted/10.1103/PhysRevE.101.023304)
- [2] [Noé, Frank, et al. "Boltzmann generators: Sampling equilibrium states of many-body systems with deep learning." Science 365.6457 (2019): eaaw1147.](https://www.science.org/doi/10.1126/science.aaw1147)
- [3][Albergo, Michael S., Gurtej Kanwar, and Phiala E. Shanahan. "Flow-based generative models for Markov chain Monte Carlo in lattice field theory." Physical Review D 100.3 (2019): 034515.](https://journals.aps.org/prd/abstract/10.1103/PhysRevD.100.034515)
- [4][Kanwar, Gurtej, et al. "Equivariant flow-based sampling for lattice gauge theory." Physical Review Letters 125.12 (2020): 121601.](https://link.aps.org/pdf/10.1103/PhysRevLett.125.121601)
- [5] [Nicoli, Kim A., et al. "Estimation of thermodynamic observables in lattice field theories with deep generative models." Physical review letters 126.3 (2021): 032001.](https://link.aps.org/pdf/10.1103/PhysRevLett.126.032001)
- [6][Neal, Radford M. "Annealed importance sampling." Statistics and computing 11 (2001): 125-139.](https://arxiv.org/abs/physics/9803008)
- [7] [Midgley, Laurence Illing, et al. "Flow annealed importance sampling bootstrap." arXiv preprint arXiv:2208.01893 (2022).](https://arxiv.org/pdf/2208.01893)
- [8] [Wu, Hao, Jonas Köhler, and Frank Noé. "Stochastic normalizing flows." Advances in Neural Information Processing Systems 33 (2020): 5933-5944.](https://proceedings.neurips.cc/paper/2020/hash/41d80bfc327ef980528426fc810a6d7a-Abstract.html)
- [9] [Caselle, Michele, et al. "Stochastic normalizing flows as non-equilibrium transformations." Journal of High Energy Physics 2022.7 (2022): 1-31.](https://arxiv.org/pdf/2201.08862.pdf)
- [10] [Nicoli, Kim A., et al. "Detecting and Mitigating Mode-Collapse for Flow-based Sampling of Lattice Field Theories." arXiv preprint arXiv:2302.14082 (2023).](https://arxiv.org/pdf/2302.14082)
- [11] [Falkner, Sebastian, et al. "Conditioning normalizing flows for rare event sampling." arXiv preprint arXiv:2207.14530 (2022).](https://arxiv.org/pdf/2207.14530.pdf)

---

### Official Review · Reviewer_9Yao · 2023-11-01

**Soundness:** 3 good
**Presentation:** 3 good
**Contribution:** 2 fair
**Rating:** 5
**Confidence:** 3

**Summary:**

The paper proposes to use normalizing flows to sample rare events. The neural networks learn the proposal distribution for the importance sampling and then use importance sampling to estimate the rare event probability. The numerical experiments show that the proposed method uses fewer function calls and has smaller errors in the average of the estimation.

**Strengths:**

1. The motivation and the problem statement are clear. The paper is also easy to follow.
2. The implementation details about the algorithm are well-explained and the math of the method is also well-written.
3. The numerical section shows experiments with synthetic data and real-world data with multiple dimensions. The paper also compares the proposed method with five other baselines.

**Weaknesses:**

1. The experiments only contain up to dimension 62, and the paper does not explain why sampling rare events at this dimension is difficult. How the comparison may look like if we compare the method with traditional sampling methods, like metropolis sampling.
2. The method's speedup and precision improvement are not clear from the languages used in the text.
3. The experiments in Figure 2 and Figure 3 look unrelated to rare event sampling but show the effectiveness of the method approximating a given distribution. It will be beneficial to get more ideas on what these figures tell us.

**Questions:**

1. Does the number of anchors matter in your experiments?
2. How do you determine the training is complete?
3. For Tables 1 and 2, do you have the measurement of time in seconds? When you say function call, does it always take the same time for different methods? If the numbers include the time of training the neural networks, would the proposed method still be faster than other methods, especially non-ML methods?
4. It would also be useful to see the confidence interval from the 20 estimations. Do you have them?